# Effect of Magnesium Hydroxide and Aluminum Hydroxide on the Thermal Stability, Latent Heat and Flammability Properties of Paraffin/HDPE Phase Change Blends

**DOI:** 10.3390/polym12010180

**Published:** 2020-01-09

**Authors:** Ru Zhou, Zhuang Ming, Jiapeng He, Yanming Ding, Juncheng Jiang

**Affiliations:** 1Jiangsu Key Laboratory of Urban and Industrial Safety, College of Safety Science and Engineering, Nanjing Tech University, Nanjing 211816, China; jcjiang@njtech.edu.cn; 2College of Urban Construction, Nanjing Tech University, Nanjing 211816, China; 201761101570@njtech.edu.cn (Z.M.); hjphlyll@njtech.edu.cn (J.H.); 3Faculty of Engineering, China University of Geosciences, Wuhan 430074, China; dingym@cug.edu.cn

**Keywords:** phase change material, thermal stability, flammability properties, metal hydroxide

## Abstract

In this study, paraffin was selected as the phase change material (PCM) and high-density polyethylene (HDPE) as the supporting material to prepare a flame-retardant PCM system. The system consisted of paraffin, HDPE, expanded graphite (EG), magnesium hydroxide (MH) and aluminum hydroxide (ATH). The thermal stability and flame retardancy were studied by thermo-gravimetric analysis (TGA), scanning electron microscopy (SEM) and cone calorimeter test (CONE). The SEM proved that the addition of MH and ATH can produce an oxide film on the surface of the composite material and form a “physical barrier” with the char layer, generated by the expansion of EG, preventing the transfer of heat and oxygen. The TGA test showed that, compared with other flame-retardant systems, the materials with added MH and ATH have a higher thermal stability and carbonization ability, and the amount of char residue has increased from 17.6% to 32.9%, which reduces the fire risk of the material. The flame retardant effect is obvious. In addition, the addition of MH and ATH has no significant effect on the phase transition temperature and latent heat value of PCMs. The CONE data further confirmed that MH and ATH can work with EG to prevent heat release, reduce the total heat release rate (THR) value and effectively suppress the generation of smoke, CO and CO_2_. The peak heat release rate (PHRR) value also decreased, from 1570.2 kW/m^2^ to 655.9 kW/m^2^.

## 1. Introduction

Phase change materials (PCMs) are substances that change the physical phase state and can absorb and release energy with the change of external temperature. Thus, PCMs are widely used in building, industry, construction and solar heating fields based on their new energy-saving, environmental-friendly properties. Especially in recent years, researchers have applied PCMs to the field of building thermal energy storage [1,2,3]. Paraffin, as one of the typical PCMs, has been widely used for building thermal energy storage, due to its large latent heat, chemical stability, low cost and wide range of commercial availability [4,5,6]. However, paraffin is easy to leak when phase change occurs, which brings inconvenience to practical applications, and so research into form-stable PCMs has emerged. The form-stable PCM is composed of a PCM and other supporting material. When the phase change occurs, the supporting material with a higher melting point will keep the materials in their original appearance, and the PCM will not leak out [7,8]. High-density polyethylene (HDPE)—as a representative of olefin polymer support materials, its molecular chain is in a crosslinked form—is favored by researchers because of its excellent heat and cold resistance, chemical stability and three-dimensional network structure for encapsulating paraffin [9]. In 1997, Inaba first melt-blended HDPE with paraffin (solid–solid phase transition temperature is 35.3 °C, the melting point is 54.1 °C) to prepare form-stable PCMs. The material contains about 74% paraffin, and the latent heat value is about 121 J/g [10]. However, due to the molecular structure of paraffin and HDPE, form-stable PCMs are extremely flammable, posing a potential fire hazard to the building. For example, the apartment fire in Jing’an District, Shanghai and the new CCTV site fire in Beijing were both typical building insulation PCM fires that caused significant losses. How to balance energy saving and fire safety is now becoming a research hotspot in this field.

In order to reduce the risk of fire hazard, the application of flame retardants has attracted more and more attention. Recent studies have shown that expandable graphite (EG) has good thermal conductivity and flame retardancy [11,12,13,14,15]. The flame-retardant effect of a single EG is limited, so we have focused on forming a synergistic flame retardant with other flame retardants, such as halogen, phosphorus and so on. However, the halogen flame retardants can release toxic substances during the combustion process. To solve this problem, inorganic flame retardants, such as metal hydroxides, have begun to replace halogen flame retardants. Metal hydroxide flame retardants have multiple functions, such as smoke elimination and flame retardancy. When the material burns, the dehydration of the metal hydroxide generates water vapor, which can effectively exert its gas phase for the flame-retardant effect of diluting the combustible gas concentration. At the same time, the generated metal oxide also plays a role as a solid phase flame retardant, effectively improving the flame-retardant performance of composite materials. Currently, the most widely used metal hydroxide flame retardants in the industry are magnesium hydroxide (MH) and aluminum hydroxide (ATH) [16]. The latest research shows that the combination of ATH and MH can significantly improve the flame retardancy of ethylene-vinyl acetate (EVA) materials. When ATH/MH is 2/1, the combined effect of the flame-retardant EVA is at its best. The limiting oxygen index value of EVA/ATH/MH composites increased from 18.3% to 34.3%, and the vertical combustion test reached V-2 level [17]. However, the above stuidies are not enough to reveal the flame-retardant mechanism of EG working with MH and ATH, so the aim of the current paper is to explore its pyrolysis behaviors and to obtain the appropriate flame-retardant mechanism.

Some studies have investigated the flammability of shape-stable PCMs based on the synergistic effects of EG with flame retardant materials such as chlorinated paraffin (CP)/antimony trioxide (AT) and ammonium polyphosphate (APP). Cai reported that the peak heat release rate of PCM4 with 21wt% APP and 4wt% EG was reduced compared to PCM2 (25wt% APP). The improved flame retardant of PCM4 is attributed to the synergistic effect between EG and APP. APP is used as an acid source, which will decompose to generate polyphosphoric acid with a strong dehydration effect during heating. The polyphosphoric acid is involved in the dehydration of EG, producing carbonaceous and phospho-carbonaceous residues, which serve as “physical barriers” to prevent heat and oxygen transfer [18]. Zhang found that EG can improve the thermal stability of paraffin/HDPE/CP/AT composites and increase the char residue at high temperatures. Adding EG to the paraffin/HDPE/CP/AT system can significantly improve the flame retardant efficiency of CP/AT [19]. Pongphat used a Brabender plastograph to prepare a fire-retardant form-stable PCM. The materials used were based on HDPE and different fire retardants such as ATH, MH, EG, APP, pentaerythrotol (PER), and treated montmorillonite (MMT). The results from the vertical burning test and TGA have shown that the form-stable PCM which contained APP+PER+MMT and APP+EG showed the best improvement in fire retardancy and thermal stability [20]. The latest research shows the synergistic flame-retardant effect and mechanism of antimony trioxide (Sb_2_O_3_) in EVA/MH system. The results show that when the total amount of flame retardant MH/Sb_2_O_3_ is 57%, the EVA/MH/Sb_2_O_3_ composite material passes the UL94 V-0 level and the limiting oxygen index value reaches 33.5%. Compared with EVA/MH composites, EVA/MH/Sb_2_O_3_ composites have better thermal stability, higher char residue, and better flame retardant [21]. Cai used montmorillonite (OMT) together with microencapsulated red phosphorus (MRP) and MH (heat absorber). They indicated that the results from both TGA and cone calorimeter have shown a significant improvement in the thermal stability and fire retardancy of the samples, which suggested that there is a synergistic effect between MRP, MH and OMT [22]. However, none of the above studies were conducted on the flame-retardant principle of MH and ATH cooperating with EG.

In order to explore the flame-retardant mechanism of MH and ATH cooperating with EG on composite PCMs in this article, HDPE was chosen as the supporting material, paraffin as the PCM, and EG, MH and ATH as flame-retardant additives. The flame retardant system of HDPE/paraffin composite PCM was prepared. The purpose of the article is to explore the flame retardancy and thermal stability of paraffin/HDPE/EG/MH/ATH systems, and to provide some theoretical support for such materials in the field of building fire protection.

## 2. Materials and Methods

### 2.1. Materials

Paraffin was available commercially with a melting temperature between 58–60 °C. It came from Wanqing Instrument Company (Nanjing, China). HDPE was purchased from Hongkai Plastic Materials Business Department (Dongguan, China). It had a melting point range of 120–130 °C. Antioxidant 1010 was supplied by Yousuo Chemical Company (Linyi, China). MH, ATH and EG were all provided by Longyue Filter Distributor (Zhengzhou, China). The expansion rate of EG was 180 mL/g. Thermal properties of the materials are shown in Table 1.

### 2.2. Preparation of Expanded Graphite

The expandable graphite was dried in a vacuum oven at 60 °C for 5 h. The dried expandable graphite was placed in a 900 °C high-temperature furnace and expanded for 60 s to obtain EG.

### 2.3. Preparation of Blend and Composite Samples

Firstly, MH, ATH, HDPE and EG were dried in a vacuum oven at 80 °C for 5 h. All the samples (Table 2) were prepared by a melt mixing process using a CTR-300 Torque Rheometer at 170 °C and 30 rpm for 15 min. The antioxidant 1010 was added to prevent the thermal-oxidative decomposition of paraffin and HDPE during heating. Then, they were hot-pressed at 150 °C for 6 min under 10 MPa pressure using a hydraulic melt press to form 100 mm × 100 mm × 3 mm samples for the cone calorimeter tests. The preparation process diagram of the PCMs is shown in Figure 1.

### 2.4. Sample Analysis

#### 2.4.1. Scanning Electron Microscopy

The morphology of PCMs was observed by a Zeiss-evo18 (Oberkochen, Germany) field emission scanning electron microscope at an accelerating voltage of 20 kV.

#### 2.4.2. Thermal Stability Test

The TGA test of samples (5–10 mg) was carried out using TG-DSC 449F3 (NETZSCH Instrument, Selb, Germany) by heating the samples from 30 to 600 °C with a linear heating rate of 10 °C/min under a nitrogen flow of 20 mL/min.

#### 2.4.3. Differential Scanning Calorimeter

The differential scanning calorimeter (DSC) analyses were done in a TG-DSC 449F3 (flow rate 20 mL/min). Samples of mass 3–5 mg were sealed in aluminum pans and heated from 30 to 200 °C at a heating rate of 10 °C/min.

#### 2.4.4. Cone Calorimeter

During the experiments, the bottom and sides of each sample were wrapped with aluminum foil, and the samples were horizontally located 25 mm away from the cone heater [23]. Samples of 100 mm × 100 mm × 3 mm were measured by a FTT0007 cone calorimeter (FTT Company, Rochester, NY, USA) under a heat flux of 50 kW/m^2^ according to the ISO 5660-1 standard. The samples were ignited by an electric spark. The general data such as heat release rate, carbon monoxide and carbon dioxide yields were reported to assess the flame retardant property of the PCMs.

## 3. Result and Discussion

### 3.1. Char Residue Morphology by SEM

The morphological structure of the char residue is shown in Figure 2. The samples are from the CONE test of PCMs (PCM2, PCM5 and PCM6 were chosen). We know that when EG is added to the polymer alone, although a thick expanded char layer is formed, the char layer is loose and porous [24], as shown in Figure 2a. As can be seen from Figure 2b,c, when MH and ATH are added to the flame retardant system, the internal structure of the material becomes more uniform and dense. Therefore, it has better heat insulation and is a better oxygen barrier, which is beneficial to the improvement of its flame-retardant performance. This is due to the decomposition reaction of MH and ATH during the heating process. Decomposed magnesium oxide and alumina cover the surface of the material and form a “physical barrier” with the expanded char layer, which not only prevents the transfer of oxygen and heat but also delays the degradation rate of the PCMs. Compared with PCM2, the flame retardancy of PCM5 and PCM6 is improved. Therefore, we can conclude that MH and ATH can form a synergistic flame-retardant effect with EG, which makes PCMs have excellent flame retardancy.

### 3.2. Thermal Stability

The TGA and differential thermogravimetry (DTG) curves of the samples are shown in Figure 3 and Figure 4, respectively. Table 3 lists the mass loss temperatures and the amount of char residue at 600 °C.

From Figure 3, the TGA curve shows three mass losses. This phenomenon corresponds to the three thermal degradation peaks of the DTG curve shown in Figure 4. This means that the thermal degradation of the composite material is mainly divided into three periods. The first stage occurs at 50–70 °C, which corresponds to the melting point of paraffin. The second stage is mainly the vaporization of paraffin and the thermal decomposition of MH and ATH, and the corresponding temperature is 200–400 °C. The third stage is at 450–550 °C, which refers to the C–C bond and C–H bond in HDPE beginning to break and decompose. This is similar to the decomposition temperature of HDPE in Table 1.

As can be seen from Table 3, the mass loss temperature (*T*_1_) of the first stage is within the melting point of the paraffin. This means that a small amount of paraffin remains on the surface of the mixture, causing the paraffin to melt after reaching the melting point. For PCM1 and PCM2, the addition of EG hindered the vaporization of the paraffin molecules, so the mass loss temperature (*T*_2_) of the second stage was increased by about 10 °C. The peak temperature of PCM3-7 is similar to or slightly earlier than PCM1. The main reason for this phenomenon is that MH and ATH begin to be thermally decomposed. When the blend is heated, MH and ATH begin to decompose in the range of 220–340 °C. Water vapor is released and a surface of the oxide attachment material is formed. The formed insulation layer prevents the transmission of oxygen and heat, protecting the substrate, and has a certain flame-retardant effect. It is for this reason that the third thermal degradation peak temperature (*T*_3_) of the flame retardant added PCM2-7 is higher than that of PCM1. It can be seen from Figure 3 and Table 3 that the *T*_3_ value of PCM2-7 is about 10 °C higher than that of PCM1. This indicates that the addition of a flame retardant is helpful for improving the thermal stability.

From Figure 3 and Table 3, we can observe that the amount of char residue of PCM2 with the addition of EG was significantly higher than that of PCM1, which demonstrates the stability and flame-retardancy of EG. The amount of char residue of PCM3-7 combined with MH and ATH is more significant than that of PCM2. With the high initial decomposition temperature of MH, the best performer was PCM3, which reached 32.9%, followed by PCM4 and PCM5, reaching 30.8% and 31.6%, respectively. Under the same conditions, the higher the MH content is, the more thermal energy is required, and the thermal stability is better, resulting in a large amount of char residue. This phenomenon indicates that the addition of MH and ATH ultimately improves the thermal stability of the paraffin/HDPE form-stable PCM, enabling it to protect the matrix material better.

### 3.3. Differential Scanning Calorimeter

The phase transition temperature and the energy storage performance of form-stable PCMs were analyzed by DSC, including the transition temperature, melting temperature, and latent heat. The transition temperature is the temperature at which PCMs undergo a solid–solid phase transition (which occurs first at low temperatures), which is the first-order phase transition. The melting temperature is the temperature at which PCMs undergo a solid–liquid phase transition (which occurs at slightly higher temperatures), which is a second-order phase transition. The DSC curve is shown in Figure 5. The phase transition temperature and the latent heat value are listed in Table 4.

For pure paraffin, the peak shoulder relates to a solid–solid transition, and the main peak is associated with the melting of the crystallites [25], as shown by the DSC curve of pure paraffin in Figure 5. Paraffin is mainly composed of linear alkanes. When the external temperature reaches the transition temperature, the linear chain of paraffin rotates around the long axis to undergo a solid–solid phase change [26]. However, this endothermic process occurs quickly and is limited, so a peak shoulder is formed on the DSC curve. The first main peak is caused by the endothermic melting of paraffin at about 70 °C. It can be found from Table 4 that the transition temperature and melting temperature of paraffin in PCM1-7 are almost the same as those of pure paraffin. The deviation range is ±1.8 °C. This illustrates that the addition of MH and ATH has no significant effect on the phase transition temperature of PCMs. Therefore, the first main peak is consistent with the trend of pure paraffin. The second main peak is caused by the endothermic melting of HDPE at 120–140 °C. This is because PCM1-7 contains HDPE, and the melting point of HDPE is about 120–130 °C (Table 1). Therefore, the temperature at which the second peak of the composite PCMs occurs is not much different. It can be seen from Figure 5 that the two main peaks are caused by the endothermic melting of paraffin and HDPE, respectively, and are in independent states. This phenomenon indicates that HDPE does not undergo a phase transition in the temperature range in which the paraffin phase changes—that is, the melting temperature exceeds the temperature at which the paraffin phase changes [27]. Thus, when paraffin is coated in HDPE and undergoes a phase change, HDPE as a support material can effectively maintain its macroscopic appearance to prevent leakage for application.

The theoretical storage of PCM is calculated by the following formula (1):(1)$$\Delta H=\Delta H_{P}\times \omega_{P}$$
where ΔH is the theoretical storage of PCMs, $\Delta H_{P}$ is the phase change storage of pure paraffin, and $\omega_{P}$ is the mass fraction of paraffin in the mixture.

The calculation results are shown in Table 4. It is obvious that, compared with the theoretical value, the actual latent heat value of PCMs is reduced. This is because when the phase transition temperature is constant, the change in the latent heat of the phase change is mainly caused by the change in the phase change entropy—that is, the change in the degree of freedom of molecular motion of the PCM before and after the phase change. This limitation mainly comes from two aspects. On the one hand, the three-dimensional network structure of HDPE limits the movement of paraffin molecules. On the other hand, when inorganic filler particles are very small, they normally act as nucleation sites for the crystallization of the polymer matrix. However, larger particles that are the result of agglomeration would rather restrict polymer chain mobility and reduce the extent of crystallization. This explains the reduced melting enthalpy in this case, because large agglomerated EG particles were observed in Figure 6.

### 3.4. Flammability Properties

The cone calorimeter (CONE) is one of the most effective tools for the assessment of fire resistance. The CONE test is based on the oxygen consumption principle. The results can be used to predict the combustion behavior of materials in real fires. The parameters available from the CONE test are heat release rate (HRR), peak heat release rate (PHRR), time to ignition (TTI), and time to PHRR (*t*_PHRR_). These values are listed in Table 5.

The TTI of the composite materials was 25–40 s. As can be seen from Table 5, compared with PCM1, the TTI and *t*_PHRR_ value of PCM2-7 were extended after the addition of EG, MH and ATH. This data shows that there is a synergy between EG, MH and ATH. This synergy improves flame retardancy and delays TTI and *t*_PHRR_. There are many factors that influence TTI—uniformity, chemical composition, rate of decomposition, and the presence of impurities [28]. In this case, the TTI is primarily determined by the rate of the decomposition of the polymer. Meanwhile, EG, MH and ATH are generally degraded at higher temperatures than paraffin, and therefore the decomposition speed is slower than PCM1, causing the TTI to be extended.

The HRR is the most important parameter in fire assessment and can be used to express the intensity of the fire [29]. In general, a highly flame-retardant system shows a low PHRR. The HRR curves are shown in Figure 7. For PCM1 and PCM2, when EG is introduced into the HDPE/paraffin blend systems, the PHRR dropped from 1570.2 kW/m^2^ to 1098.2 kW/m^2^, a reduction of nearly 30%. Therefore, the addition of EG into HDPE/paraffin system enhances the barrier properties of the char layer, such that the HRR is reduced. For PCM3-7, the PHRR is further reduced compared to PCM2. PCM6 is reduced by up to 40.3%. It has been reported that the rate of char layer formation and the compactness of a char layer have a strong influence on the reduction of the HRR and the improvement in the flame retardancy. Therefore, the addition of MH and ATH increases the density of the char layer, further hinders the heat transfer and improves the flame retardancy of the mixture.

Figure 8 shows the total heat release rate (THR) for all the materials. In Table 5, THR values are listed. The THR of PCM1 and PCM2 reached 120.8 MJ/m^2^ and 109 MJ/m^2^ within 250 s, respectively. This shows that the fire of PCM2 with EG added is less than that of PCM1. These data confirm the flame-retardant effect of EG. The THR of PCM3-7 is further reduced, and the value of PCM6 is the lowest, at 103 MJ/m^2^. It is not difficult to find that the fire spread of PCM3-7 with MH and ATH is decreased. The main reason is the synergistic flame-retardant effect of MH, ATH and EG. EG expands when heated, forming a char layer, preventing the release of heat. MH and ATH absorb some of the heat to undergo a decomposition reaction, which further consumes heat, resulting in a reduction in THR. This illustrates that there is a synergistic effect between MH, ATH and EG in a paraffin/HDPE flame-retardant system, which can reduce the THR value during combustion.

Generally, the smoke production rate (SPR) plays a critical role in fire conditions. Similarly to the HRR curve, MH and ATH can significantly reduce the SPR value. The SPR curve is shown in Figure 9. The SPR value of PCM2 with EG added is slightly smaller than that of PCM1, and the smoke suppression effect is not obvious. However, for PCM3-7 with MH and ATH, it is obvious that the SPR value is much smaller than PCM1 and PCM2, and the lowest value is PCM6, achieving the best smoke suppression effect. The reason is that MH and ATH are decomposed during the heating process, and the generated water vapor dilutes the concentration of the surrounding smoke, and the surface of the generated oxide covering material isolates the transmission of oxygen, further reducing the generation of smoke. Hence, the addition of MH and ATH could effectively prevent gas transfer between the flame zone and the burning paraffin/HDPE, and retard the generation of smoke. The phenomenon also indirectly proves that MH and ATH are good smoke suppressants and have application prospects.

The concentration of carbon monoxide (CO) and carbon dioxide (CO_2_) is one of the most important parameters for assessing fire safety. CO is the main toxic gas that causes death in fires [30]. In Figure 10 and Figure 11, the CO and CO_2_ evolution are described during the burning process, respectively. They have similar trends. The total CO yields of seven samples were all very low. PCM1 and PCM6 have the highest and lowest CO and CO_2_ concentrations, respectively. Compared to PCM1, the CO and CO_2_ contents of PCM2 are significantly reduced. This is because of the addition of EG. The CO and CO_2_ content of PCM3-7 are further reduced to a safe level. This phenomenon is consistent with the results of SPR. The water vapor generated by the decomposition of MH and ATH dilutes the concentration of CO and CO_2_ around the combustibles, resulting in a decrease in the content of CO and CO_2_. The production of CO and CO_2_ is accompanied by the burning of materials, so the flame-retardant effect of the blends means to reduce both CO and CO_2_ emissions. From the results, PCM6 has the lowest CO and CO_2_ content. Because the initial decomposition temperature of ATH is lower than MH, before the temperature reaches the decomposition temperature of MH, ATH decomposition generates Al_2_O_3_ attached to the surface of the material, forming an oxide film barrier layer. The higher the ATH content is, the denser the barrier layer. Therefore, PCM6 effectively suppresses the generation of smoke, CO and CO_2_.

Among the flammability characteristics of materials, the mass loss rate (MLR) is also particularly important. In Figure 12, the MLR curves for the selected samples (PCM1, PCM2, PCM4-6) are shown. The MLR values of PCM4-6 and PCM2 were both lower than the PCM1 blend (Table 5). This means that PCM4–6 decompose slowly in a fire, and their flame-retardant effect is obvious. PCM6 has the lowest MLR value, which is 0.13 g/s. Because PCM6 contains the most ATH, the initially formed oxide film barrier layer is thick, which the heat and oxygen cannot easily get into the inside of the mixturel, resulting in a lower MLR value. The MLR data further support the flame-retardant efficiency of EG, MH and ATH, which is very important for the practical application of MH and ATH as metal hydroxide flame retardants.

## 4. Conclusions

The thermal stability, latent heat value and flammability properties of paraffin/HDPE, paraffin/HDPE/EG blends and a paraffin/HDPE/EG/MH/ATH flame retardant system were investigated. The results show that the thermal stability and flammability properties of the paraffin/HDPE/EG/MH/ATH system are significantly improved. The SEM proves that the addition of MH and ATH can produce an oxide film on the surface of the mixed paraffin/HDPE, and form a “physical barrier” with the char layer generated by the expansion of EG, preventing the transfer of heat and oxygen. The TGA test shows that, compared with other flame retardant systems, the materials with added MH and ATH have a higher thermal stability and carbonization ability, and the amount of char residue increases from 17.6% to 32.9%, which reduces the fire risk. The flame-retardant effect is obvious. In addition, the DSC data illustrates that the addition of MH and ATH has no significant effect on the phase transition temperature and latent heat value of PCMs, so the material is theoretically feasible as a heat storage material. The CONE data further confirm that MH and ATH can work with EG to prevent heat release, reduce the THR value, and the PCM6 effectively suppresses the generation of smoke, CO and CO_2_. The PHRR value also decreased from 1570.2 kW/ m^2^ to 655.9 kW/m^2^, a decrease of more than 58%. Although the char residue content of PCM3 reached 32.9%, PCM6 was 26.2%. After comprehensive data analysis, compared with other PCMs, PCM6 shows better thermal stability and flame retardancy. These results have confirmed that MH and ATH can synergize with EG to form a good flame-retardant system, which provides data support for the application of such materials in the construction field.

## Figures and Tables

**Figure 1 polymers-12-00180-f001:**
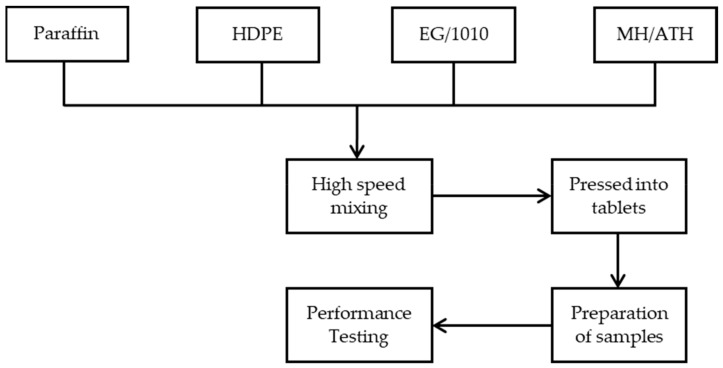
Preparation process diagram of the phase change materials (PCMs).

**Figure 2 polymers-12-00180-f002:**
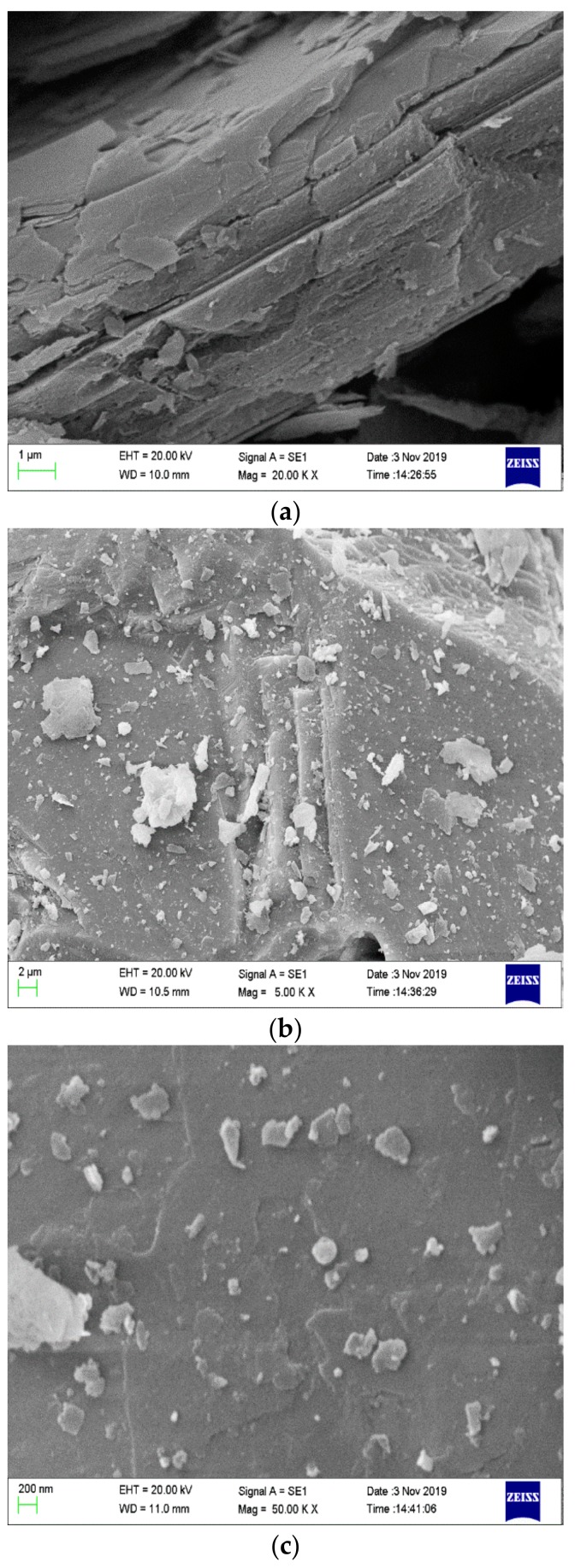
Scanning electron microscopy (SEM) photographs of char layer from PCMs after the CONE test. (**a**) PCM2, (**b**) PCM5, and (**c**) PCM6.

**Figure 3 polymers-12-00180-f003:**
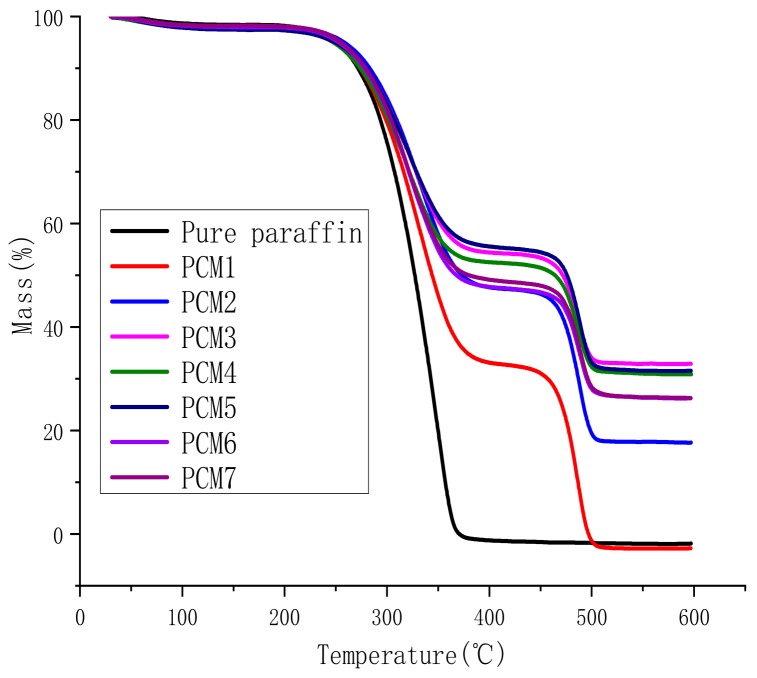
Thermo-gravimetric analysis (TGA) curves of PCMs.

**Figure 4 polymers-12-00180-f004:**
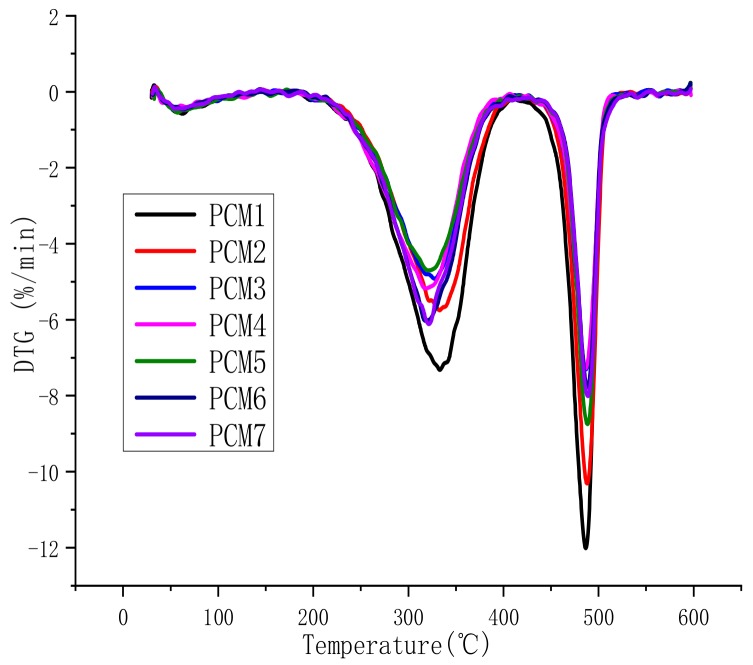
Differential thermogravimetry (DTG) curves of PCMs.

**Figure 5 polymers-12-00180-f005:**
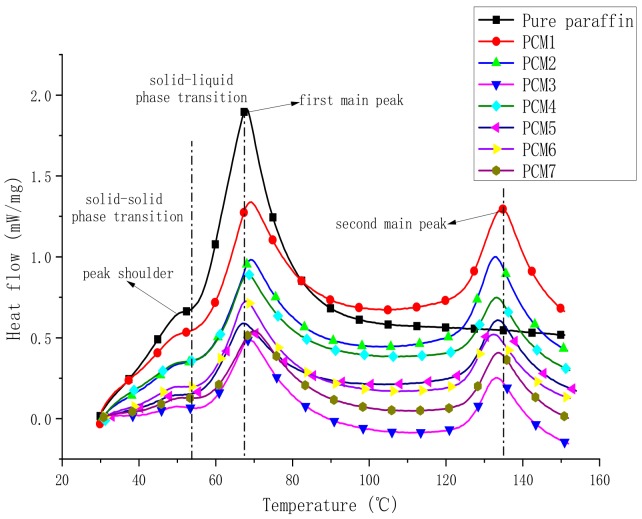
Differential scanning calorimeter (DSC) curves of the samples.

**Figure 6 polymers-12-00180-f006:**
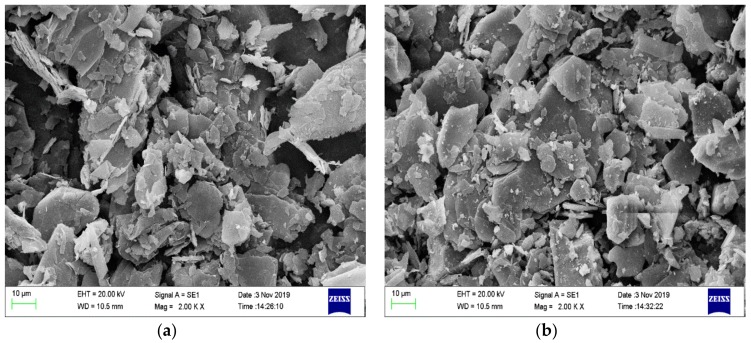
SEM photographs of PCM2 and PCM4. (**a**) Expanded graphite (EG), (**b**) EG + magnesium hydroxide (MH) + aluminum hydroxide (ATH).

**Figure 7 polymers-12-00180-f007:**
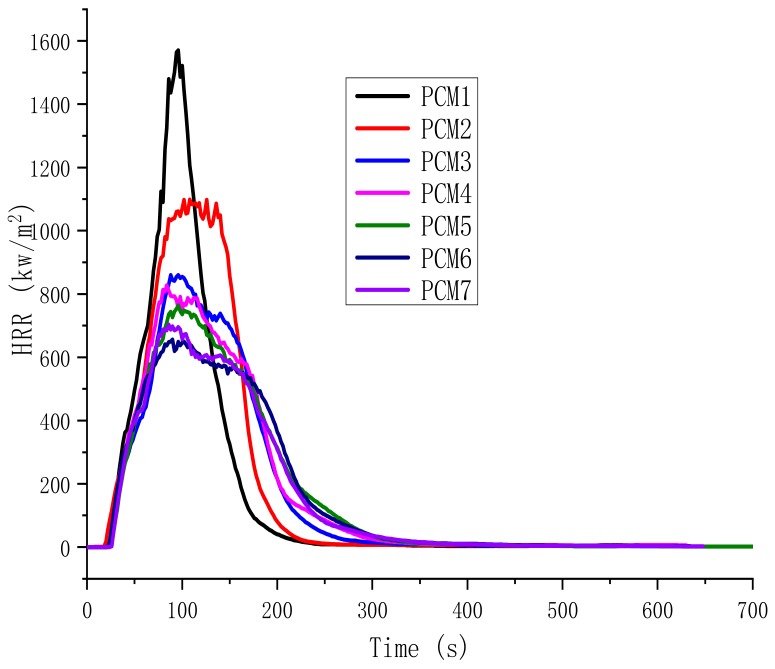
Heat release rate (HRR) curves of the PCMs.

**Figure 8 polymers-12-00180-f008:**
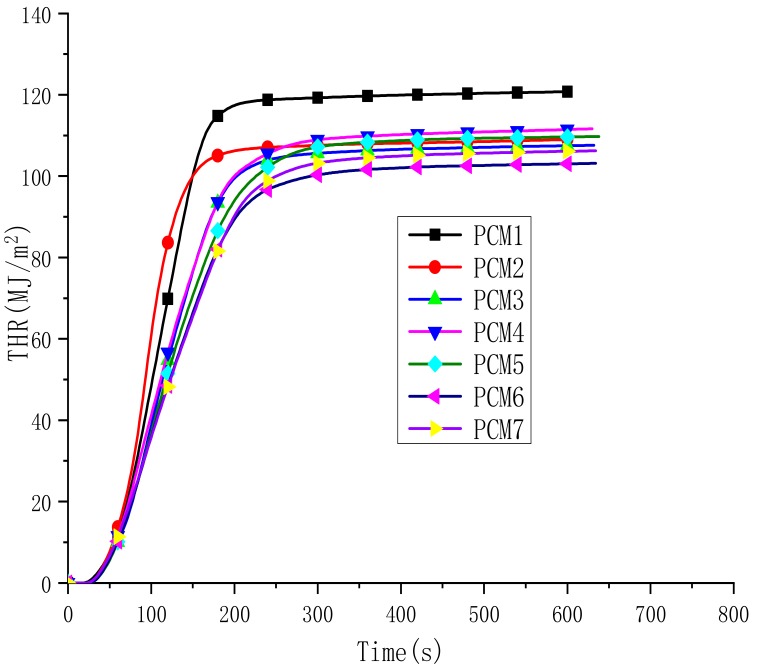
Total heat release rate (THR) curves of the PCMs.

**Figure 9 polymers-12-00180-f009:**
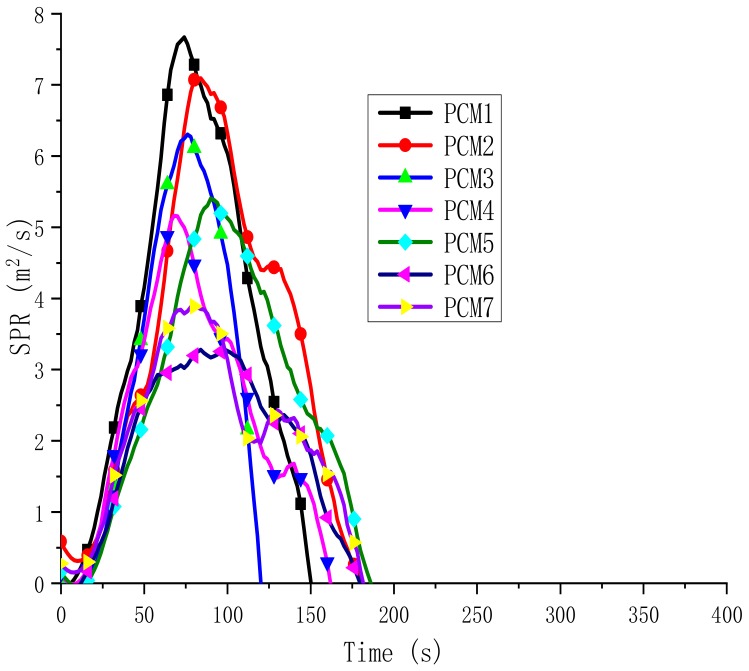
Smoke production rate (SPR) curves of the PCMs.

**Figure 10 polymers-12-00180-f010:**
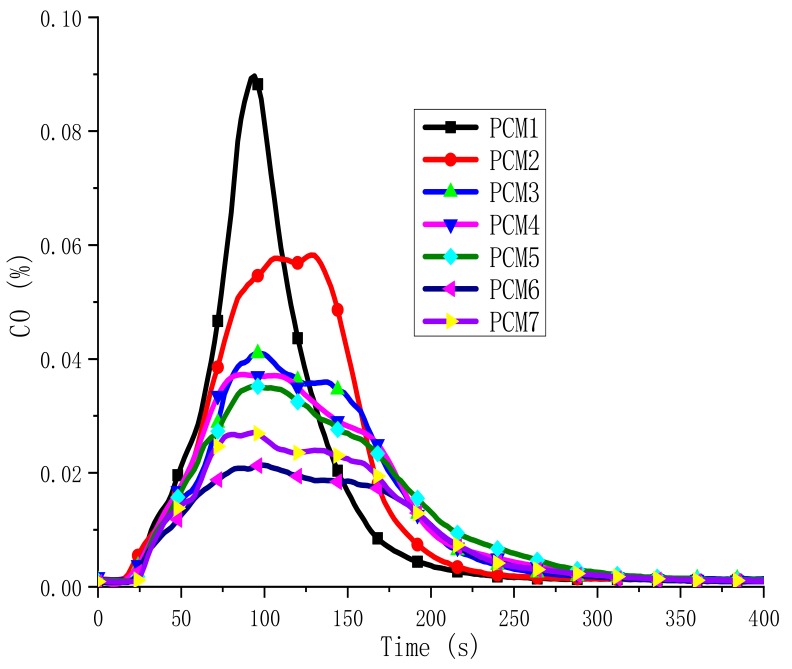
CO evolved in PCMs.

**Figure 11 polymers-12-00180-f011:**
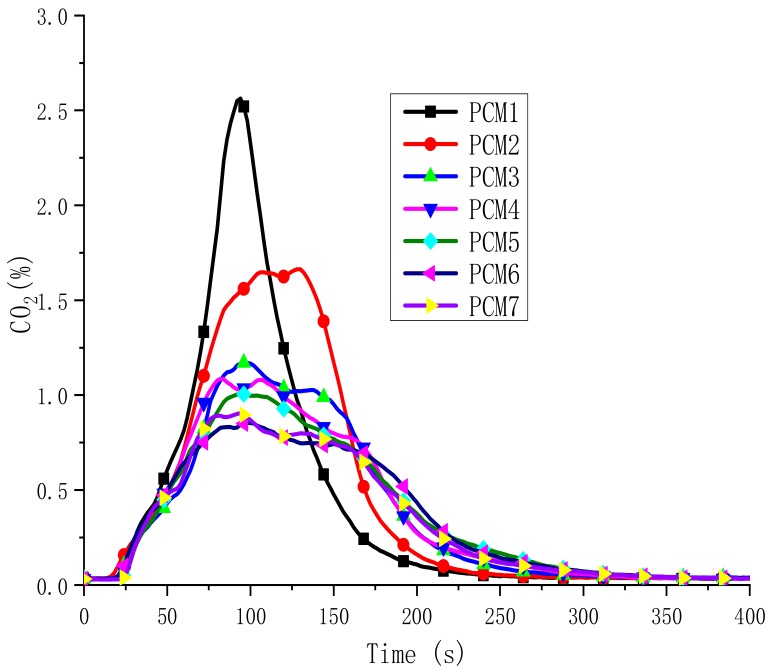
CO_2_ evolved in PCMs.

**Figure 12 polymers-12-00180-f012:**
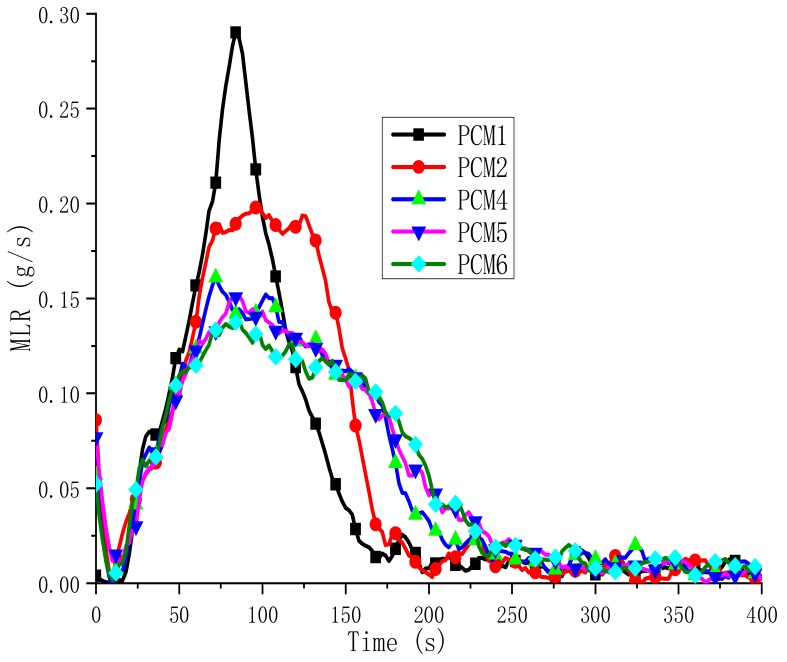
Mass loss rate (MLR) curves of the PCMs.

**Table 1 polymers-12-00180-t001:** Thermal properties of the material.

	Paraffin	HDPE	ATH	MH
**Melting Temperature/°C**	58–60	120–130	--	--
**Decomposition Temperature/°C**	--	420	220	340
**Density/g/cm^3^**	0.9	0.95	2.4	2.36
**Thermal Conductivity/W/(m·K)**	0.12	0.5	--	--

**Table 2 polymers-12-00180-t002:** Sample compositions used in this study.

Sample	^a^ Paraffin/g	^a^ HDPE/g	EG/g	ATH/g	MH/g	^b^ Antioxidant1010/g
PCM1	28	12	0	0	0	0.12
PCM2	23.8	10.2	6	0	0	0.12
PCM3	18.2	7.8	6	0	8	0.12
PCM4	18.2	7.8	6	2	6	0.12
PCM5	18.2	7.8	6	4	4	0.12
PCM6	18.2	7.8	6	6	2	0.12
PCM7	18.2	7.8	6	8	0	0.12

^a^ paraffin/HDPE = 7/3; ^b^ Antioxidant 1010 content is 0.3%.

**Table 3 polymers-12-00180-t003:** Mass loss temperatures of PCMs.

Samples	*T*_1_/°C	*T*_2_/°C	*T*_3_/°C	Char Residue/%
PCM1	62.8	323.3	476.3	0
PCM2	64.1	332.6	488.1	17.6
PCM3	63.4	327.9	486.4	32.9
PCM4	68.2	318.2	486.2	30.8
PCM5	57.6	321.2	488.3	31.6
PCM6	63.1	317.9	488.4	26.2
PCM7	62.1	321.6	488.7	26.3

**Table 4 polymers-12-00180-t004:** Thermo-physical properties of the PCMs.

Samples	Transition Temperature/°C	Melting Temperature/°C	Latent Heat/J g^−1^	Theoretical Value/J g^−1^
paraffin	50.4	68.1	93.5	-
PCM1	49.8	69.2	59.0	65.4
PCM2	50.6	69.3	41.0	55.6
PCM3	49.9	68.8	30.1	42.5
PCM4	51.2	68.6	36.7	42.5
PCM5	50.9	67.0	30.5	42.5
PCM6	50.3	68.0	35.8	42.5
PCM7	51.3	69.9	32.0	42.5

**Table 5 polymers-12-00180-t005:** Flammability data of the samples used in this study.

Sample	PHRR (kW/m^2^)	THR (MJ/m^2^)	TTI (s)	*t*_PHRR_ (s)	MLR (g/s)
PCM1	1570.2	120.8	25	*86*	0.3
PCM2	1098.2	109.0	28	108	0.2
PCM3	859.9	107.6	30	98	0.17
PCM4	827.7	111.7	31	94	0.15
PCM5	762.9	109.9	34	96	0.14
PCM6	655.9	103.0	38	90	0.13
PCM7	703.4	106.2	29	86	0.16

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
