# Peer review of "Effect of Magnesium Hydroxide and Aluminum Hydroxide on the Thermal Stability, Latent Heat and Flammability Properties of Paraffin/HDPE Phase Change Blends"

_polymers, 2020, doi:10.3390/polym12010180_

Round 1
Reviewer 1 Report
Topic: The Effect of Magnesium Hydroxide and Aluminum Hydroxide on the Thermal Stability, Latent Heat, and Flammability Properties of Paraffin/HDPE Phase Change Blends
Authors: Ru Zhou, Zhuang Ming, Jiapeng He, Yanming Ding, Juncheng Jiang
I am thankful to the kind editor to provide me this manuscript for the review process. The following questions and suggestions are mandatory to improve the quality of the study.
Improve the abstract with technical and critical information. Mention amount the PCM, HDPE, EG, ATH and MH. WHY are you referring the new composite material as shape-stable material? Since it is composite PCM. WHAT is PHRR? Explain the complete abbreviation first time. No intended problem is highlighted in introduction. Why did you conduct this study? What was the engineering problem? Address all the questions in detail. Very poor literature review. Add the latest paper relevant to current study. No critical results are mentioned from previous studied in literature review. Approve the literature review. Clearly mention the aims and objectives of current study. Clearly provide the companies links and thermal properties specifications of PCM, HDPE, EG, ATH and MH in tabulated form. Explain the preparation process of composite PCM in detail with the help of schematic diagram. Increase the size of resolution of all figures. Difficult to the observe the figures. Figure 2: Caption should be below the figure on same page. Please check carefully. Check Eq. 1. Type error. Use proper mathematical tool to write. WHY there are two peaks of all samples in DSC curves. Provide the clear scientific reasoning. Use the legend text in English. I strongly emphasize that authors should respect the International language for readers. Mention the thermophysical properties of composite materials in tabulated form such as thermal conductivity, specific heat, latent heat of fusion etc. Poor quality of all figure. Please improve the resolution. Very poor quality of the results in Figure 7. Results are not well discussed and technically presented. OVERALL the quality of the presentation is very poor. There are number of grammar and type errors. Conclusions are not well summarized and technically justified. Please address point by point.
Author Response
Dear Reviewer:
Thank you for your comments which are all valuable and very helpful for revising and improving our paper, as well as the important guiding significance to our studies. We have focused on the comments carefully and made corrections accordingly, which we hope meet with your approval.
Please see the attachment

Reviewer 2 Report
The article entitled “The Effect of Magnesium Hydroxide and Aluminum Hydroxide on the Thermal Stability, Latent Heat, and Flammability Properties of Paraffin/HDPE Phase Change Blends” has minor novelty.
There are many articles in the literature on this subject, which were not addressed. For example:
Fire retardants for phase change materials. https://doi.org/10.1016/j.apenergy.2011.02.005: „Fire-retarded form-stable phase change material (PCM) products consisting of paraffin (RT21) (or propyl ester), high density polyethylene (HDPE) and fire retardants were prepared using the Brabender Plastograph. The properties of the form-stable PCM, containing different types of fire retardants such as magnesium hydroxide, aluminium hydroxide, expanded graphite (EG), ammonium polyphosphate (APP), pentaerythritol (PER) and treated montmorillonite (MMT) were classified using vertical burning test, thermogravimetry analysis (TGA) and differential scanning calorimeter (DSC).” Magnesium Hydroxide Paraffin/HDPE: Y. B. Cai, Q. F. Wei, D. F. Shao, Y. Hu, L. Song & W. D. Gao (2009) Magnesium hydroxide and microencapsulated red phosphorus synergistic flame retardant form stable phase change materials based on HDPE/EVA/OMT nanocomposites/paraffin compounds, Journal of the Energy Institute, 82:1, 28-36, DOI: 10.1179/014426008X370988Authors pointed that the novelty of study is the investigation of MH, ATH and EG synergistic effect on fire retardancy. From this perspective, article could be accepted for publication after major revision.
Specific comments:
More professional English language should be used. Introduction is very poor. The detailed review should be done on paraffin/HDPE systems filled with used flame retardants should be described. There are a lot of articles on this subject. The idea of phase change materials should be explained in the Introduction. Practical applications should be specified. Authors mentioned them many times (from the Introduction to Conclusions) without specifying. Lines 40-47 – are lacking the citation Line 44 – what the authors mean by the “three-dimensional network structure of PE”? Figure 4 is lacking the indication of the transition type (exo or endo). Black line has a description in Chinese. All figures have poor quality. What is [J] in many citations? The term “transition temperature” should be explained. This is important because, for specialist in physical chemistry of polymers melting temperature is a first order phase transition. What is the difference between “transition temperature” and “melting temperature”? I can see that authors addressed the DSC shoulder temperature as the transition temperature and described as solid-solid transition. This phenomena should be explained from the physical chemistry point of view in the main text and supported with proper citation. Melting temperatures of HDPE were neglected in Table 3. Line 188 – it is obvious that “no chemical reaction occurs among the paraffin and HDPE”. It is obvious that sometimes two components are chemically bonded and they represent their own characteristic transitions. Different explanation should be found and supported with proper citation. The articles dealing with parafine/PE miscibility should be helpful. Line 186 – authors refer to Table 2 instead Table 3. Lines 191 – 193 “Thus, when paraffin is coated in HDPE and undergoes a phase change, HDPE as a support material can effectively maintain its macroscopic appearance to prevent leakage for application.” This finding is well known and used for the manufacturing of HDPE filled with microencapsulated paraffin, such as in the work: https://doi.org/10.1016/j.enconman.2014.06.061. In this study, being currently under review, a simple mixing was done. Line 297-298 – Conclusions – “In addition, the flame retardant mixed paraffin/HDPE system has a lower phase transition temperature and can be used as a potential heat storage material.” In Table 2, it is indicated that phase transition temperature was almost unaltered. Conclusion should be completely remodeled in order to summarize result supporting the novelty, indicated in the Introduction.Author Response
Dear Reviewer:
Thank you for your comments which are all valuable and very helpful for revising and improving our paper, as well as the important guiding significance to our studies. We have focused on the comments carefully and made corrections accordingly, which we hope meet with your approval.
Please see the attachment.

Reviewer 3 Report
The study of Zhou et al. deals with trials to render a phase change material (PCM) flame retardant. A flame retardant mixture comprising of expanded graphite (EG) and aluminum trihydroxide (ATH) and magnesium dihydroxide (MDH), respectively, was incorporated into PCM consisting of paraffin and high-density polyethylene (HDPE).
The thermal properties of samples containing 15 wt% EG and different loadings of the metal hydroxides were investigated by different scanning calorimetry (DSC) and thermo-gravimetric analysis (TGA) and their burning behaviors were investigated by cone calorimetry. In addition, the morphologies of residues remaining after cone calorimetric tests were evaluated by SEM. Net PCM samples and PCM samples only containing EG were investigated for comparison (the burning behavior of net paraffin was investigated as well).
The introduction part as well as the section 2 (materials and methods) provide all information necessary for the reader of the paper. In section 3.2, page 5, the thermal degradation of the samples (TGA curves) is discussed. However, the following argumentation seems to me implausible (line 148-149): ” …The third stage was at 450-550°C, which referred to thermal-oxidative decomposition of HDPE”. Please take into consideration that the TGA curves were recorded under nitrogen atmosphere! Therefore, the authors should urgently answer that question: How can a thermal oxidation occur under nitrogen atmosphere where no oxygen is present?? An error has crept in in Fig. 4 that has to be corrected: please replace the Chinese word by an English one (paraffin).
Investigations by cone calorimetry revealed that combinations of EG with ATH showed better flame retardant effect compared to EG alone. Interestingly, partial replacing of ATH by MDH showed an amplifying effect on the flame retardant action (due to formation of more dense char residue). An advantage of that flame retardant mixture is that it diminish the smoke release, especially the evolution of carbon monoxide. However, a disadvantage is the rather high overall loading applied (35 wt%). In consequence, the application of large amount of flame retardant additives seriously decreased the latent heat (deteriorated the ability to store energy) of PCM samples. The means the flame retardant PCM described in the presented study is rather not suitable for practice.
Nevertheless, the manuscript provides interesting results so that I recommend it for publication in the journal polymers. However, the criticism mentioned above should be considered.
Author Response
Dear Reviewer:
Thank you for your comments which are all valuable and very helpful for revising and improving our paper, as well as the important guiding significance to our studies. We have focused on the comments carefully and made corrections accordingly, which we hope meet with your approval.
Please see the attachment.

Round 2
Reviewer 1 Report
Topic: The Effect of Magnesium Hydroxide and Aluminum Hydroxide on the Thermal Stability, Latent Heat, and Flammability Properties of Paraffin/HDPE Phase Change Blends
Authors: Ru Zhou, Zhuang Ming, Jiapeng He, Yanming Ding, Juncheng Jiang
I am thankful to the kind editor to provide me this manuscript for the review process.
Authors have presented all the queries from my side and all the information provided in the manuscript is fulfilling the journal criteria, so I recommend accepting this paper for the journal of Polymers.